# Bronchoscopy and Thermal Ablation: A Review Article

Aristides J. Armas Villalba *  and Bruce F. Sabath

Department of Pulmonary Medicine, The University of Texas MD Anderson Cancer Center,
1400 Pressler St. Unit 1462, Houston, TX 77030, USA; bsabath@mdanderson.org
* Correspondence: ajarmas@mdanderson.org

**Abstract:** Thermal ablative techniques are part of the armamentarium of interventional pulmonologists for the treatment of a diverse range of pathologies, but most importantly used in airway obstruction and airway bleeding. These techniques can be categorized based on their onset of action into rapid and delayed ablative methods. Understanding the nuances of each technique is essential, as most clinical scenarios demand a combination of modalities, commonly referred to as a "multi-modality approach". This comprehensive review aims to elucidate the fundamental principles of rapid ablative techniques, including laser therapy, argon plasma coagulation (APC), and electrocautery, along with the research that underpins their clinical application.

**Keywords:** bronchoscopy; interventional pulmonology; APC; laser; electrocautery; airway bleeding; airway obstruction





## 1. Introduction

The utilization of thermal therapies finds its historical roots in antiquity, as evidenced by the writings of Hippocrates. In his seminal work "Aphorisms", Hippocrates articulated a fundamental medical principle "what drugs will not cure, the knife will; what the knife will not cure, the cautery will; what the cautery will not cure must be considered incurable" [1]. This ancient aphorism encapsulates the enduring recognition of the therapeutic potential inherent in thermal treatments within the annals of medical history.

As human civilization has advanced through history, the field of medicine has advanced tremendously, driven in large part by the march of technological progress. Within this contemporary era, thermal therapies have assumed a prominent role within the ever-expanding repertoire of interventional pulmonologists, representing a therapeutic modality frequently employed in the care of our patient population.

The two most common indications to use thermal ablative techniques are airway obstruction with endoluminal disease and bleeding.

The planning and implementation of thermal ablation procedures entail a meticulous assessment of several pivotal factors. These considerations encompass the bronchoscope of choice, the clinical presentation of the patient, the fraction of inspired oxygen tolerated by the patient and the risk of airway fire, and the presence of airway prothesis.

In the context of addressing central airway obstruction, the prevailing clinical approach typically involves the utilization of multiple instruments and techniques during the intervention, a practice commonly referred to as "multimodality". Hence, the scarcity of randomized controlled trials comparing head-to-head the different tools available.

We learned that patients that are very symptomatic and with a poor functional status, benefit the most from therapeutic interventions in central airway obstruction [2]. As opposed to delayed techniques (e.g., cryotherapy or photodynamic therapy), rapid ablative techniques confer immediate effect with rapid improvement of a patient's symptoms.

In patients undergoing treatment cancer treatment, it is of paramount importance improve the performance status since it can alter the treatment options (or withdrawal) available.

This review will focus on the use of rapid ablative techniques: laser, argon plasma coagulation (APC), and electrocautery.

To treat our patients appropriately, the team must be composed of multidisciplinary proceduralists that have been trained to the highest standards. It is known that each tool has a variable learning curve, and each proceduralist will perform variably while acquiring new abilities. Hence the importance of training under experienced physicians, along with implementation of simulation models to further optimize our performance [3–6].

## 2. Laser Therapy

Laser's acronym is "light amplification of stimulated emission of radiation". It is delivered with the use of fiber optics via rigid or flexible bronchoscopy and is exclusively used for the treatment of endobronchial disease.

There are multiple laser systems available (Table 1), each with their own characteristics that makes them unique depending on coagulation and vaporization needs. The laser that is most used and widely available for bronchoscopic use is the neodymium:yttrium–aluminum–garnet (Nd:YAG) laser [4].

**Table 1.** Comparison of the different laser characteristics.

| Characteristics | Nd:YAG | $CO_2$ | Diode | Argon |
|---|---|---|---|---|
| Cutting | + | +++ | + | + |
| Coagulation | +++ | + | ++ | ++ |
| Tissue depth (mm) | 6–8 | 0.1–0.3 | 1–2 | 1–2 |

The characteristics were ranked from lowest (one +) to highest (three +++) degree of cutting or coagulation.

The power density, ratio of absorption, and scattering will be the biggest determinants of tissue effect [5]. Distance to target will influence the power density, and the relationship is inversely proportional (as the distance to target shortens, the power density increases).

When the goal is to achieve coagulation, low power settings are used, the probe-to-target distance is maintained around 1 cm, and the operator will note how the tissue turns white ("blanching effect") after activating the laser.

On the other hand, if the power is increased or the fiber is closer to the target, the tissue will be carbonized and vaporized, with the operator noting charring and blackening of the tumor [6].

When the use of laser is contemplated, it is of utmost importance to have a deep understating of the patient's anatomy, clinical presentation, and imaging studies, as well as the technique to be utilized.

### 2.1. Technique

Ideally, the patient will be under general anesthesia, to minimize risks and complications, with a $FiO_2$ at the moment of activation of 40% and below to decrease the possibility of airway fire.

Classically, the power setting for Nd:YAG laser is variable, most commonly between 20 and 100 Watts and the activation time goes from 0.2 s to 2 s depending on the clinical scenario [7].

The fiber must be parallel to the airway to avoid cutting the airway and misfiring critical blood vessels or other anatomic structures. The direction of the laser must be always controlled and organized to avoid unnecessary damage to surrounding structures.

Once the tissue effects are seen, settings can be adjusted accordingly, as well as the distance to target and the type of fiber (contact vs non-contact). Special attention must be taken to always keep the tip of the fiber clean and clear of debris to guarantee an accurate laser activation.

After the tissue is treated and bleeding is controlled, other techniques can supplement tissue debulking/excision, like mechanical debulk or other thermal ablative techniques, as part of the multimodal approach to obtain better results.

*2.2. Outcomes*

As mentioned before, the research supporting laser by itself is very limited since the vast majority of the studies were conducted using a multimodal approach (Table 2).

**Table 2.** Evidence available for the use of laser therapy.

| Authors | Type of Research | Sample Size | Outcomes | Comments |
| --- | --- | --- | --- | --- |
| Cavaliere et al. [8,9] | Retrospective analysis | 1000 patients initially, but a subsequent analysis included 2× sample size | Improved airway lumen in 92% cases. The other 8% were driven by extrinsic compression. Low complication rates. | Favored the use of rigid bronchoscope Nd:YAG laser Change in symptoms post-procedure were not addressed |
| Squiers et al. [10] | Retrospective analysis | 99 patients, 261 procedures | Symptom improvement was observed in 90% of benign cases and 77% of malignant etiology. Proximal lesions had better outcomes than distal disease. Low rate of complications and mortality (<1%). | Holmium:YAG laser was utilized in 100% of the cases |
| Perin et al. [11] | Retrospective analysis | 464 patients | Overall complication rate was 8.4%. Identified age (>60 years), current smokers, COPD, hypertension, and heart disease as significant risks factors associated with complications. | Only patients with central airway obstruction were included |

One of the largest studies supporting the use of laser was conducted by Dr. Cavaliere et al. in Italy [8], a retrospective analysis of a thousand patients (1396 applications) that underwent laser application over a period of 5 years. The paper supports the use of laser for benign and malignant disease, with an adequate safety profile for the patient.

Approximately 92% of the procedures were performed via rigid bronchoscopy and general anesthesia. Reviewing the outcomes for airway obstruction, 92% of the time airway patency was improved by the end of the case, and the other 8% were unsuccessful due to a significant component of extrinsic compression and/or distal disease present.

Interestingly, the authors mention that right after using the laser, there was improvement in ventilation, but it is unclear if that translates into improvement of symptoms for the patient since it was not clearly measured. Centrally located tumors had better outcomes; the authors describe that central airway obstruction (trachea and main bronchus) achieved excellent results in more than 95% of the patients.

In terms of adverse events, the main complications were hypoxemia and those related to general anesthesia. Only 10/1396 applications were complicated with severe hemorrhage (defined as more than 250 mL) but were managed without difficulties. Two cases of pneumomediastinum that resolved spontaneously and four pneumothoraces. A very low mortality rate (0.35%) was associated with the procedure.

Later in 1996, Dr Cavaliere published a subsequent manuscript [9] with twice the number of patients and confirmed similar outcomes for patients that underwent laser application.

Alternative uses for the laser have been proposed [10,11], including treatment of endobronchial tumors with curative intent. Unfortunately, these research projects are mostly limited by small samples and retrospective analyses. Hence, surgical resection remains the gold standard for patients with early disease and appropriate clinical condition.

### 3. Argon Plasma Coagulation

Argon Plasma Coagulation (APC) is a thermal ablative method that belongs to the electrosurgical techniques along with the electrocautery.

In contrast to electrocautery, APC has been described as a non-contact technique that utilizes argon to induce coagulation and or tissue debridement.

The APC can be used with the flexible or rigid bronchoscope with the aid of a probe that will deliver the argon. Once the probe is activated with the pedal, the gas flows from the probe; once it is in contact with the electric current, the gas ionizes (plasma) and transfers electrons to the surrounding tissue at a very high frequency. This process causes a thermal (heat) reaction with subsequent cellular damage.

The gas diffuses to surrounding areas with less electrical resistance that is translated into a uniform but superficial tissue effect (around 3 mm depth). As opposed to laser therapy, APC can successfully treat broad areas that are not necessarily in front (zero degrees) of the probe (e.g., tangential, radial, or around anatomic corners) [12].

APC is considered simple, safe, and very affordable, and is used very frequently with other methods to achieve the desired outcome for the patient.

In the United States, the probes are available in three different sizes (outer diameter): 1.5, 2.3, and 3.2 mm. This is important knowledge since it will influence the bronchoscope to use given the size of the working channel.

#### 3.1. Technique

The technique involves a deep understanding of the patient, clinical condition, anatomy, and the goal of the procedure, and always being mindful of the $FiO_2$ required since the risk of airway fire increases when the $FiO_2$ is above 40%.

Before the procedure starts, a grounding pad must be placed in the patient to avoid shocking the patient. The probe must be chosen based on the target characteristics (location, size/extension, presence of bleeding) and the size of the working channel of the bronchoscope.

To avoid damage to the bronchoscope, the probe is inserted into the working channel until the first black mark is seen. The probe is approximated to the area of interest without making contact, the pedal is activated, and the tissue is treated accordingly.

As mentioned before, APC is a non-contact method, where the probe is approximated to the target and then activated. If the probe is impaled in the tissue, the risk of gas embolism increases.

There are three modes available in terms of gas flow: Forced (continuous), Pulsed (interrupted), and Precise (continuous). The proceduralist selects the mode (commonly Forced), power (20–40 Watts), and flow (0.3 to 1.8 L/min).

#### 3.2. Outcomes

APC has been proven over the years to be a safe tool for the interventional pulmonologist when correctly used [13]. Complications rates are low, but risks include gas embolism and airway perforation [14], especially when the probe is activated for prolonged periods of time and/or is very close/inside the tissue.

The largest retrospective was published by Dr. Reichle and colleagues [15], where they describe 482 procedures in 364 patients with very low complication rates (3.7% per procedure). The most common scenario for APC use was malignant airway obstruction in 51% of the cases, achieving good results in 67% of the patients. The second scenario was airway bleeding, and hemostasis was achieved in 99% of the cases.

Dr. Morice [16] also published his experience with 60 patients, but only for hemostasis and airway obstruction. In terms of bleeding, hemostasis was achieved in 100% of the cases.

With the airway obstruction, patients had significant improvements in lumen size as well as symptom relief. Interestingly, no complications were reported in the manuscript.

Table 3 compares the size of the APC probes with the bronchoscope working channel. Table 4 summarizes the evidence that supports the use of APC in airway obstruction and bleeding.

**Table 3.** Bronchoscope working channel width and APC probe sizes.

|  | Bronchoscope Working Channel | APC Probe Sizes |
| --- | --- | --- |
| BF-P190 | 2 | 1.5 |
| BF-1TH190 | 2.8 | 2.3 |
| BF-XT190 | 3.2 | 3.2 |

All measurements are in millimeters.

**Table 4.** Evidence available for the use of APC.

| Authors | Type of Research | Sample size | Outcomes | Comments |
| --- | --- | --- | --- | --- |
| Reichle et al. [15] | Retrospective | 364 patients with 482 procedures | Successful treatment of malignant central airway obstruction in 67% of the patients. Bleeding controlled in 99% of the cases where APC was utilized. Very low complication rates (3.7% per case). | Main indication to use APC was airway obstruction, followed by airway bleeding. |
| Morice et al. [16] | Retrospective | 60 patients | Hemostasis was achieved in 100% of the cases. Symptom improvement and airway lumen improved in cases of airway obstruction. | No complications reported. |
| Reddy et al. [17] | Case series | 3 patients | Three patients developed life threating gas embolism while using APC bronchoscopically. The gas was activated for short periods of time (1–3 s) | Rigid bronchoscopy. Single center study with large referral system. |

## 4. Electrocautery

As opposed to APC, electrocautery encompasses a group of techniques that are contact-based, meaning the tool must touch the tissue in order to cause a local effect. Multiple tools are available for the bronchoscopist: snare, hot forceps, knife, and the blunt probe.

The principle of electrocautery is explained by the flow of high-frequency electrical current that generates heat and causes tissue destruction. Amperage and voltage are important variables that will determine the tissue effect.

Coagulation is achieved by setting the generator to high amperage with low voltage, cutting with low amperage and high voltage. By balancing the amperage and voltage, a blending effect can be obtained [18].

In comparison to laser, electrocautery is cheaper and equally safe when used by an experienced operator.

While preparing for the procedure, special attention must be paid to patients with cardiac devices since the electrical current can be misinterpreted by the device and misfire, leading to potentially life-threating arrhythmias.

### 4.1. Technique

The technique varies, and will depend on the tool to be used. As with the other thermal ablative techniques, the $FiO_2$ must be reduced to 40% or less before the pedal is activated.

Always make sure the patient has a grounding pad placed to avoid an electrical discharge to the patient.

In general, the electrocautery generator is activated with two pedals of different colors to ease the identification and functionality. The blue pedal is configured for coagulation (low voltage) and the yellow pedal is for cutting (high voltage).

Typical settings configure the power between 20–60 Watts; usually coagulation mode is on the lower range and cutting on the higher end [19]. Ideally, the proceduralist should start with the lowest setting, and increase it as needed based on the intraprocedural findings with the current energy.

The tool to be used is determined by the type of lesion that needs to be treated. If there is a polypoid or pedunculated mass, the snare is the tool of choice.

The snare is available in multiple sizes and can be used with the flexible or rigid bronchoscope, but handling of the snare is easier via flexible bronchoscopy. The stalk or narrowest part of the mass needs to be identified.

The snare is inserted through the working channel and a short segment of the sheath is exposed. The assistant slowly pushes the snare out, and the idea is to loop the mass and slowly start closing the snare by the stalk or narrowest point. Once closed and tension is felt, the initial activation must be with the blue pedal (coagulation), followed by cutting to minimize bleeding until the stalk is resected.

Once resected, the mass can be removed with suction or the preferred tool (e.g., forceps).

A blunt electrocautery probe is frequently used to treat bleeding and debulk. As a contact method, the probe is advanced until contact is made with the lesion and the pedal is activated for short periods of time until the desired local effect is desired. The longer the pedal is activated, the higher damage and tissue penetration is achieved.

Another tool is the hot forceps, with a theoretical benefit of giving the ability to obtain a tissue biopsy with lower incidence of bleeding, but in clinical practice that is not always the case [14]. Nowadays, cold forceps are rarely used given that they have few benefits over their hot counterpart.

The electrocautery knife is needle-shaped and widely utilized to treat benign stenotic lesions with the goal of radial cuts and later balloon dilation. It is important to do multiple short-lived activations given how powerful the needle knife is and the risk of airway perforation.

### 4.2. Outcomes

Modalities under the umbrella of electrocautery are very cheap and safe. Most of the research combines multiple techniques, and the vast majority of the data are retrospective. The safety concerns include airway fire, airway perforation, airway bleeding, and skin burns to the patient if the grounding pad is not utilized.

Dr. Wahidi and collaborators [20] describe their experience with 94 patients (117 procedures), where endoscopic improvement was achieved in 94% of the cases. Of these patients, 71% had symptomatic improvement as well, correlating with the improvement in radiologic studies (specifically CT Scans) as well with low rates of major complications (0.8%).

One of the few prospective studies evaluating the safety of endobronchial electrosurgery was conducted in Japan [21]. A total of 37 patients were enrolled (54 procedures) and 35 of them experienced satisfactory outcomes. In one patient (1/35), the slow bleeding was not controlled at the end of the case, and a second patient had a calcified polypoid lesion (metastatic osteosarcoma) that could not be excised with the snare.

Out of the 37 patients, 26 had respiratory symptoms (dyspnea, hemoptysis, and/or inability to expectorate sputum) and 23 reported improvement of the symptoms after the procedure was completed.

### 5. Conclusions

Thermal ablative techniques hold a pivotal position in the therapeutic armamentarium for addressing a spectrum of endobronchial diseases as well as managing airway bleeding. The proficient mastery and comprehensive understanding of the available instrumentation are of paramount significance for the proceduralist to execute interventions with precision and efficacy, thereby ensuring optimal patient care.

Throughout the course of the procedure, the judicious selection and utilization of multiple tools are anticipated and warranted, contingent upon the imperative need to meet the specific requirements of the patient. The decision-making process is closely tied to several determinants, including the proceduralist's level of comfort and expertise, the anatomical considerations and proximity to adjacent structures, as well as the clinical presentation of the patient.

The three strategies delineated (Table 5) within the scope of this article exhibit distinct sets of advantages and disadvantages, albeit sharing certain commonalities in terms of their technical requirements.

**Table 5.** Summary and general considerations.

| Method | Mechanism of Action | Tissue Penetration | Advantages | Disadvantages | Special Considerations | Complications |
|---|---|---|---|---|---|---|
| Laser | The light energy emitted by the fiber transforms into heat, causing tissue damage. Tissue effect is influenced by system settings and distance to target. | Up to 8 mm (Nd:YAG laser). Non-contact method. | Can be used with flexible and/or rigid bronchoscope. Easier to aim and control. The same tool can cut and coagulate by modifying the settings. | Very expensive equipment. Variable tissue penetration. Limited effect in dark tissue due to poor light absorption. | Low power and distance from target will favor coagulation. High power and short distance to target favors carbonization and vaporization of tissue. | Hypoxemia. Airway perforation (pneumomediastinum and pneumothorax). Bleeding. Very low mortality. |
| APC | Argon is activated with electric current in order to transfer electrons to surrounding tissue to induce coagulation and/or tissue devitalization. | Approx 3 mm. Non-contact method. | Very affordable. Can be used to stop bleeding and tumor debulking as well. Can treat areas that are difficult to reach with other methods. | Risk of gas embolism. If severe bleeding, can be difficult to control since the gas will be in contact with blood and not with the tissue per se. Can interfere with pacemakers. | A safe distance must be kept from the tissue and vessels in order to avoid the risk of gas embolism. | Gas embolism. Airway perforation. Airway fire. |
| Electrocautery | A probe is activated with electric current, and once in contact with the tissue will cause the desired effect. | Tool dependent. Contact method. | Cheapest of the three methods. Multiple tools available to use depending on the need. Two pedals are used: cutting and coagulation. Contact method. | Highly related to the tool utilized. Can interfere with pacemakers. | | Airway perforation. Bleeding. Airway fire. Skin burns if patient is not grounded. |

As emerging technologies continue to evolve, it becomes crucial to remain up-to-date with the latest literature available inherent to each technique. In this journey, the patient's well-being is our north star.

**Author Contributions:** Conception and design: A.J.A.V. and B.F.S.; provision of study materials or patients: A.J.A.V. and B.F.S.; collection and assembly of data: A.J.A.V. and B.F.S.; data analysis and interpretation: A.J.A.V. and B.F.S.; manuscript writing: A.J.A.V. and B.F.S. All authors have read and agreed to the published version of the manuscript.

**Funding:** This research received no external funding.

**Data Availability Statement:** No new data were created or analyzed in this study. Data sharing is not applicable to this article.

**Conflicts of Interest:** Both authors have no conflict of interest to disclose.

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
