# Peer review of "Bronchoscopy and Thermal Ablation: A Review Article"

_2673-527X, doi:10.3390/jor4010003_

Round 1
Reviewer 1 Report
Comments and Suggestions for Authors
The article present the thermal ablative techniques as part of multimodality approach of airway obstruction and airway bleeding. The topic is very important in interventional pulmonology field and the authors present the fundamental principles of these rapid ablative techniques, a meticulous assessment of several factors that influences the success of the methods and some results and outcomes from the literature. There are only few data from the literature about the IP techniques especially prospective studies , so the references are appropriate.
Nice work. Congratulations.
Author Response
Dear Reviewer
I really appreciate your feedback in the construction of this manuscript.
Best regards,
Aristides J. Armas Villalba, M.D
Reviewer 2 Report
Comments and Suggestions for Authors
Recently, electrocautery devices have some different modes such as soft-coag. Are there any comments on it?
Author Response
Dear Reviewer
I appreciate your constructive feedback for the creation of this manuscript. The different modes that you are mentioning refers to change in prepopulated equipment settings and the different tools available.
In the new revision, I have included part of those changes and hope the message is clearer.
Again, I am grateful for your input.
Warm regards,
Aristides J. Armas Villalba
Reviewer 3 Report
Comments and Suggestions for Authors
Dear Authors,
I read the "Bronchoscopy and Thermal ablation: a Review Article" paper.
Although you propose a comprehensive literature review, the manuscript needs to be more conclusive.
Finding the in-depth revision of the current knowledge you promised the reader takes work. Although the manuscript is well written, it needs to have relevance.
1) You must define the revision proposal you are attempting for. If you define your job as a comprehensive review, you must cover all the knowledge for each group of techniques you mentioned. The paper must be improved and expanded.
2) Tables are very useful. If you prefer to put the information discursively, refer to big tables where all the main papers are mentioned.
3) No references or comments on the learning curves, skill practice, or technical difficulties characterize the three ablative techniques. There are differences in the pros and cons of flexible and rigid bronchoscopy.
4) The conclusion paragraph is scarce. Notify the reader about other techniques driving different treatments (e.g., cryoablation).
Author Response
Dear Reviewer
I hope you are doing well. First, I would like to say thank you for the thoughtful feedback you provided me with and I agree with your suggestions.
Based on your comments, I have changed the draft and hope it delivers a better message to the audience.
Again, thank you for your constructive feedback.
Warm regards,
Aristides J. Armas Villalba
Round 2
Reviewer 3 Report
Comments and Suggestions for Authors
Dear Authors,
Thank you for your effort in ameliorating the manuscript.
Ok, for me.